# Increased VA-ECMO Pump Speed Reduces Left Atrial Pressure: Insights from a Novel Biventricular Heart Model

**DOI:** 10.3390/bioengineering12030237

**Published:** 2025-02-26

**Authors:** Anirudhan Kasavaraj, Christian Said, Laurence Antony Boss, Gabriel Matus Vazquez, Michael Stevens, Jacky Jiang, Audrey Adji, Christopher Hayward, Pankaj Jain

**Affiliations:** 1Faculty of Medicine, UNSW Sydney, Kensington 2052, Australia; 2Mechanical Circulatory Support Laboratory, St Vincent’s Centre for Applied Medical Research, Darlinghurst 2010, Australia; 3Faculty of Engineering, UNSW Sydney, Kensington 2052, Australia; 4Graduate School of Biomedical Engineering, UNSW Canberra, Canberra 2600, Australia; 5Department of Cardiology, St Vincent’s Hospital Sydney, Darlinghurst 2010, Australia; 6Department of Cardiology, Royal Prince Alfred Hospital, Camperdown 2050, Australia; 7Sydney Medical School, The University of Sydney, Camperdown 2050, Australia

**Keywords:** heart failure, cardiogenic shock, mechanical circulatory support, extracorporeal membrane oxygenation, physiology, hemodynamics

## Abstract

Background and aims: The effect of veno-arterial extracorporeal membrane oxygenation (VA-ECMO) on left atrial pressure (LAP) in the presence of interventricular interaction and the Frank–Starling mechanism is unknown. We developed and validated a mock circulatory loop (MCL) incorporating a novel, 3D-printed biventricular heart model and Frank–Starling algorithm, and used this model to assess the determinants of LAP during VA-ECMO support. Methods: The MCL was designed to allow a separate ventricle or biventricular configuration, with or without an active Frank–Starling mechanism. The biventricular model with Frank–Starling mechanism was validated in terms of (1) the presence and degree of ventricular interactions; (2) its ability to simulate Frank–Starling physiology; and (3) its capacity to simulate normal and pathological cardiac states. In the separate ventricle and biventricular with Frank–Starling models, we assessed the effect on LAP of changes in mean aortic pressure (mAoP), ECMO pump speed, LV contractility and ECMO return flow direction. Results: In the biventricular configuration, clamping RA inflow decreased RAP, with a concurrent decrease in LAP, consistent with direct ventricular interaction. With a programmed Frank–Starling mechanism, decreasing RAP was associated with a significant reduction in both LV outflow and LV end-systolic pressure. In the biventricular model with a Frank–Starling algorithm, the MCL was able to reproduce pre-defined normal and pathological cardiac output, and arterial and ventricular pressures. Increasing aortic pressure caused a linear increase in LAP in the separate ventricle model, which was attenuated in the biventricular model with Frank–Starling mechanism. Increasing ECMO pump speed caused no change in LAP in the separate ventricle model (*p* = 0.75), but significantly decreased LAP in the biventricular model with Frank–Starling mechanism (*p* = 0.039), with stabilization of LAP at the highest pump speeds. Changing the direction of VA-ECMO return flow did not affect LAP in either the separate ventricle (*p* = 0.91) or biventricular model with Frank–Starling mechanism (*p* = 0.76). Conclusions: Interventricular interactions and the Frank–Starling mechanism can be simulated in a physical, biventricular MCL. In their presence, the effects of VA-ECMO on LAP are mitigated, with LAP reduction and stabilization at maximal VA-ECMO speeds.

## 1. Background and Aims

Cardiogenic shock (CS) is characterized by hypoperfusion and hypotension despite sufficient or elevated filling pressures [1]. The main causes of CS are acute myocardial infarction and decompensated heart failure [1,2,3]. Increasingly, CS is treated using temporary mechanical circulatory support (MCS), with veno-arterial extracorporeal membrane oxygenation (VA-ECMO) being used commonly and across an increasing number of sites [4,5,6].

VA-ECMO can provide full cardiorespiratory support in cardiac arrest and/or CS due to a range of underlying causes, including acute myocardial infarction, acute decompensated heart failure, myocarditis, refractory arrhythmia, peri-partum cardiomyopathy, and post-cardiotomy [6,7]. VA-ECMO can be positioned peripherally, extracting vena caval or right atrial blood, and most commonly returning oxygenated blood to the femoral artery, with retrograde flow to the vital organs [6,8,9]. Less commonly, it is positioned centrally, extracting right atrial blood, and returning it to the proximal aorta after oxygenation, with antegrade flow to the vital organs [8]. However, as V-A ECMO returns oxygenated blood to an arterial inflow, it can increase left ventricular (LV) afterload, impairing aortic ejection and causing accumulation of blood within the LV cavity [10]. This can cause LV distension, elevated LV end-diastolic pressure (LVEDP), left atrial pressure (LAP) and pulmonary capillary pressure, causing pulmonary edema [7,8,10].

Our understanding of the mechanisms by which these effects occur is incomplete. In particular, it is unclear whether the increased afterload during VA-ECMO is a flow-mediated or purely pressure-mediated effect. A recent study performed in mock circulatory loop (MCL) suggested that arterial pressure and LV contractility, but not VA-ECMO flow or direction (central vs. peripheral), influence LAP during VA-ECMO support [11]. In addition, it is not clear whether the secondary increase in left-sided filling pressures is mitigated by interventricular interactions or reflex mechanisms such as Frank–Starling. Finally, the existing strategies to prevent these effects, including LV venting surgically, with retrograde aortic pump placement or interatrial septostomy, are either costly, invasive or unproven in terms of efficacy. Better understanding and predicting the deleterious effects of VA-ECMO is critical, not only with regards to device selection in CS, and also in determining whether to use VA-ECMO for upfront or provisional cardiorespiratory support during high-risk interventional procedures [12].

We sought to address these knowledge gaps using a novel MCL incorporating a biventricular heart model. We first validated the biventricular MCL in terms of its ability to model interventricular interactions and the Frank–Starling mechanism, and its ability to reproduce healthy and cardiogenic shock physiology. We then investigated the effect of VA-ECMO pump speed on LAP, and the effect on this relationship of changes in mean aortic pressure (mAoP), LV contractility, ECMO return flow direction, and the presence or absence of interventricular interactions and a Frank–Starling mechanism.

## 2. Methods

### 2.1. Mock Circulatory Loop Design

The MCL used for this study has been previously described, although only incorporating a ‘separate-ventricle’ heart model [11]. In both the separate ventricle and novel biventricular models, ventricles were pneumatically compressed and expanded using air movement driven by a proprietary electro-pneumatic driver to move a glycerol solution through the MCL. Specific MCL components, including the novel biventricular heart model, are described below and outlined in Figure 1A,B.

#### 2.1.1. Separate Ventricle Model

The pre-existing separate ventricle heart model had two silicon diaphragms (77 mm × 49 mm) in transparent sealed air chambers (170 mm × 95 mm × 95 mm) simulating the right ventricle (RV) and left ventricle (LV). Four mechanical bi-leaflet valves (27 mm On-X Life Technologies Inc. Heart Valve Product Group, Austin, TX, USA) functioned as ventricular inflow and outflow valves. LV and RV contractility were changed by altering the voltage delivered to electropneumatic regulators (SMC Corporation, Tokyo, Japan), which scaled the driving pressure delivered to each ventricle using compressed air.

#### 2.1.2. Biventricular Model

A dual-chambered silicon diaphragm (112 mm × 72 mm × 81 mm) was placed in a detachable, sealed air chamber (228 mm × 154.5 mm × 191 mm) to simulate a biventricular heart with an RV, LV, an interventricular septum, and a pericardium (Figure 2, Appendix A). St Jude Medical Epic 25mm (ESP100-25A, Abbott Cardiovascular, Abbott Park, IL, USA), St Jude Medical Biocor 25mm (B100-25A, Abbott Cardiovascular, Abbott Park, IL, USA) and St Jude Medical Trifecta 25mm (TFGT-25A & TF-25A, Abbott Cardiovascular, Abbott Park, IL, USA) artificial valves were used in the biventricular heart.

#### 2.1.3. Frank–Starling Mechanism

The algorithm used to program the Frank–Starling mechanism used right atrial pressure (RAP) and LAP to modify RV end-systolic pressure (RVESP) and LV end-systolic pressure (LVESP), respectively (Appendix A). RVESP and LVESP were measured using Millar pressure transducers. The algorithm was designed and coded into Simulink (MathWorks, Inc., Natick, MA, USA). The relationship between atrial pressure and ESP was linear, analogous to the end-systolic pressure–volume relationship (ESPVR). Contractility was altered by adjusting either the slope or intercept of this linear relationship.

#### 2.1.4. Circulatory/Vascular Components

The MCL had systemic and pulmonary circulations. Polyvinyl chloride (PVC) tygon tubing (Saint-Gobain Corp, Courbevoie, France) was used to simulate ‘blood vessels’ with expansile and resistance properties. Resistance could be further modulated through adjusting tubing clamps present on different MCL sections. The systemic circulation had an upper body and a lower body section, each with individual resistances, which contributed to the overall systemic vascular resistance (SVR). The pulmonary circulation consisted of pulmonary arterial and pulmonary venous sections and had pulmonary vascular resistance (PVR). Aortic pressure (AoP), pulmonary arterial pressure (PAP), LAP and RAP were measured using fluid-filled pressure transducers.

Aortic, systemic venous, pulmonary arterial, and pulmonary venous compliances were modelled using four identical, hermetically sealed, cylindrical Windkessel chambers (100 mm diameter × 320 mm height). The four chambers contained water at different levels, which could be altered using a syringe attached to a three-way tap atop each chamber to change the compliance for arterial and venous MCL sections.

Systemic flow was measured using a BioProTT Clamp-On Transducer (em-tec GmbH, Finning, Germany) placed proximal to the right atrium. The MCL was filled with a 40% *w*/*w* glycerol/water solution. To simulate a hematocrit of 42–44%, the solution was maintained between 28 and 30 degrees Celsius by immersing part of the lower limb circulation in a Water Bath (Grant Instruments Ltd., Cambridge, UK).

#### 2.1.5. ECMO Components

The MCL simulated an ECMO pump using a fixed speed centrifugal pump (HeartWare HVAD, Medtronic, Minneapolis, MN, USA). The HVAD pump speed could be altered from 1800 RPM to 4000 RPM. The ECMO inflow limb drew fluid from the right atrium. It had a retrograde outflow to the femoral artery (a section of tubing in the lower limb circulation), so that ECMO outflow entered the systemic circulation in a retrograde direction, as well as an antegrade outflow to the proximal aorta where ECMO outflow entered the systemic circulation in an antegrade direction. ECMO flow was measured using a BioProTT Clamp-On Transducer.

### 2.2. Experimental Protocol

#### 2.2.1. Validation of Biventricular Heart Model

The MCL was validated in three domains: the presence of direct ventricular interaction (VI); the presence of a demonstrable Frank–Starling mechanism; and its ability to reproduce normal and pathological hemodynamic states.

Ventricular interactions

The presence of direct VI was assessed in both the pre-existing separate ventricle model, and in the novel biventricular model, through clamping the RA inflow and assessing the immediate effect on mean RAP and LAP.

Frank–Starling mechanism

In the biventricular model, the presence of an ‘intrinsic’ Frank–Starling mechanism was assessed through clamping of RA inflow to reduce preload and assessing the effect on LVESP and LV outflow. This was then repeated following the installation of a dedicated Frank–Starling algorithm, described above.

Normal and pathological cardiac states

In the biventricular model with Frank–Starling algorithm, compliances and ventricular contractilities were altered in various combinations to replicate four cardiac states: normal, left ventricular failure (LVF), right ventricular failure (RVF), and biventricular failure (BVF), as outlined in Table 1.

#### 2.2.2. Effect of VA-ECMO on Left Atrial Pressure in Biventricular MCL

All compliances, systemic and pulmonary vascular resistance, and ventricular contractilities were altered in various combinations to replicate three pathological cardiac states: left ventricular failure (LVF), right ventricular failure (RVF), and biventricular failure (BVF). In each cardiac state, the effect on LAP of alteration in one or more of the following variables was assessed: cardiac state, ECMO orientation, ECMO speed, mean aortic pressure and LV contractility. The specific variables assessed in each experiment are summarized in Appendix A.

### 2.3. Data Acquisition and Analysis

The Millar pressure control unit and the fluid pressure transducers were zeroed before running the MCL. All experiments measured mean aortic pressure (mAoP), left atrial pressure (LAP), right atrial pressure (RAP), mean pulmonary arterial pressure (mPAP), ECMO flow and systemic flow. LV and RV pressures were also measured for experiments conducted in the BV and BVFS models. All measurements were monitored by a custom interface created on ControlDesk (dSpace GmbH, Paderborn, Germany), and pump speed was recorded from the HVAD pump controller interface. All recordings were carried out under steady state conditions.

Mean pressures and flows were extracted from the ECMO experiments in the SV, BV and BVFS models to ascertain determinants of LAP. Simple linear regression was conducted to assess the relationship between variables, one-way analysis of variance (ANOVA) was used to assess changes in outcome variables between different states, and multiple linear regression was used to assess the different models and predictors of LAP. A *p*-value < 0.05 was considered statistically significant for all data analysis.

## 3. Results

### 3.1. Validation of Biventricular MCL


*Ventricular interdependence*


In the separate ventricle model, clamping the RA inflow caused a minor increase, followed by a decrease in RAP, and a slow increase in LAP (Figure 3A). In the biventricular model, clamping the RA inflow caused a slight increase, followed by a decrease in RAP, and a concurrent decrease in LAP (Figure 3B). The ratio of LAP change to RAP change in the BV model was 0.47.


*Frank–Starling mechanism*


Testing for an intrinsic Frank–Starling mechanism in the biventricular model indicated that decreasing RAP as a result of clamping the RA inflow was associated with a significant decrease in LV outflow but only a minor reduction in LVESP (Figure 4A). Following introduction of the programmed Frank–Starling mechanism in the biventricular model, decreasing RAP was associated with a significant reduction in both LV outflow and LVESP (Figure 4B).


*Reproduction of normal and diseased cardiac states*


In the biventricular model with Frank–Starling mechanism, the four cardiac states outlined in Table 1 were successfully reproduced. The specific values obtained are summarized in Table 2.

### 3.2. Determinants of Left Atrial Pressure During V-A ECMO

#### 3.2.1. Effect of Aortic Pressure on Left Atrial Pressure

Increasing aortic pressure while maintaining constant LV contractility and ECMO pump speed was associated with a linear increase in LAP for all ventricular failure states in the separate ventricle model (Figure 5A). This effect was attenuated in the biventricular model with Frank–Starling mechanism, with no observed increase in LAP despite increased mAoP (Figure 5B).

#### 3.2.2. Effect of ECMO Pump Speed on Left Atrial Pressure

Increasing ECMO pump speed while holding mAoP constant and LV contractility constant was not associated with any change in LAP in the SV model (F(1, 12) = 0.106, *p* = 0.75, Figure 5C) or the BV model (F(1, 12) = 2.541, *p* = 0.14). In the BVFS model, however, increasing ECMO pump speed was associated with significantly decreased LAP (F(1, 12) = 5.363, *p* = 0.039, Figure 5D).

#### 3.2.3. Effect of Changes in LV Contractility and Pump Speed on Left Atrial Pressure

In the separate ventricle model, decreasing LV contractility while maintaining a constant SVR and ECMO pump speed was associated with linearly increased LAP and linearly decreased mAoP (Figure 5E). This effect was observed across all ECMO pump speeds, with higher pump speeds producing linearly higher LAP.

In the biventricular model with Frank–Starling mechanism, decreasing LV contractility while maintaining a constant SVR and ECMO pump speed was similarly associated with increased LAP and decreased mAoP at lower pump speeds (Figure 5F). At pump speeds of 3400 RPM and higher, these effects were minimized, with almost no effect from altering contractility on either LAP or mAoP. LAP was lowest at the maximum pump speed (4000 RPM).

#### 3.2.4. Effect of ECMO Direction on Left Atrial Pressure

There was no statistically significant difference in LAP between an antegrade and retrograde ECMO outflow in either the separate ventricle (F(1, 12) = 0.015, *p* = 0.91) (Figure 5G) or biventricular model with Frank–Starling mechanism (F(1, 12) = 0.1, *p* = 0.757) (Figure 5H).

## 4. Discussion

Cardiogenic shock complicates 5–8% of ST-elevation myocardial infarction (STEMI) cases and 2.5% of non-STEMI cases [1,3], and is associated with high in-hospital mortality rates [13]. While VA-ECMO remains a mainstay of CS treatment, the associated increased afterload—thought to be at least in part due to retrograde arterial return flow—may result in deleterious effects, including LV distension, elevated filling pressures and pulmonary edema. This study validates a novel physical model replicating two key aspects of human circulation, and applies this model to improve our understanding of the VA-ECMO-related effects. Our key findings were as follows. (1) Biventricular interactions mitigate the effect of arterial pressure on LAP; (2) increased VA-ECMO pump speed is associated with *reduced* LAP in the setting of Frank–Starling mechanism; (3) in a biventricular model with Frank–Starling mechanism, high VA-ECMO pump speed stabilizes LAP at a low value, regardless of underlying LV contractility; and (4) direction of VA-ECMO return flow does not affect LAP. Together, these findings add significant granularity to our understanding of the ECMO–heart–circulation interaction and challenge the current paradigm regarding this interaction.

### 4.1. Biventricular Interactions Mitigate the Effects of Arterial Pressure on LAP

In the separate ventricle model, increasing aortic pressure in the presence of constant VA-ECMO pump speed caused increased LAP, while increasing VA-ECMO pump speed in the presence of constant aortic pressure did not affect LAP. These findings are consistent with previously published data [11], and confirm that the effects of VA-ECMO on LAP are driven by changes in arterial pressure, rather than by retrograde ECMO flow. However, in the biventricular model with Frank–Starling mechanism, the impact of arterial pressure on LAP was mitigated, with no change in LAP despite increased arterial pressure in the presence of constant pump speed. This is likely due to interventricular interactions, as the increased LV filling pressure, caused by increased mAoP, is transmitted to the RV via the interventricular septum. Further, rightward shift of the interventricular septum may reduce RV stroke volume, thereby reducing LV filling and further attenuating the increase in LAP.

### 4.2. Increased ECMO Speed Is Associated with Reduced LAP in the Presence of a Frank–Starling Mechanism

In the biventricular model with Frank–Starling mechanism, increasing ECMO speed *decreased* LAP. This finding is contrary to the current clinical paradigm, in which an increased ECMO pump speed, particularly during decreased LV contractility, contributes to increased LAP and can cause or worsen pulmonary edema [14]. This observed decrease in LAP with increased ECMO speed may be due to direct reduction in right-sided filling pressure due to drainage into the ECMO circuit, causing both a reduction in RV contraction and reduced left-sided filling volume, as well as directly transmitted reduction in left ventricular filling pressure via interventricular interactions [15].

### 4.3. LAP and mAoP Are Stabilized at High ECMO Speed in a Biventricular Model with Frank–Starling Mechanism, Regardless of LV Contractility

In the separate ventricle model, decreased LV contractility was associated with higher LAP at all ECMO speeds, and higher ECMO speed was associated with both higher mAoP and higher LAP. However, in the biventricular model with Frank–Starling mechanism, these relationships only held at lower ECMO pump speed; at higher speeds, both mAoP and LAP stabilized, and the effect of LV contractility was abolished. Furthermore, in this model, there was a *reduction* in LAP seen at the highest speed, regardless of LV contractility. This effect is likely due to the greater effect of VA-ECMO in reducing RA pressure at high pump speeds, with transmission of lower right-sided filling pressure to the left chambers through direct interventricular interactions and indirectly via reduced RV contraction and pulmonary flow.

### 4.4. ECMO Direction Does Not Affect LAP

Both the separate ventricle and biventricular models demonstrated no change in LAP between an antegrade and retrograde ECMO outflow insertion, indicating that increased afterload from VA-ECMO is not influenced by direction of flow. This challenges the prevailing understanding that the increased LV afterload and LAP is caused by retrograde VA-ECMO flow. Instead, it highlights that VA-ECMO exerts its effect on LAP via complex interactions between pump speed, LV contractility and mAoP, and hence the importance of managing these parameters in ECMO-supported CS, rather than simply decreasing ECMO pump speed or flow [16]. Given the extent to which these findings challenge the current clinical paradigm, they require verification in a large animal or human model.

### 4.5. Implications and Future Directions

This research confirms previous findings regarding the role of mAoP and LV contractility in determining LAP during VA-ECMO in the separate ventricle model [11,17]. However, in the presence of biventricular interactions and physiological Frank–Starling mechanism in our study, these effects were largely mitigated. Indeed, at the highest ECMO flows, LAP remains fixed at a low level, regardless of LV contractility. These findings should give pause to the current, simplistic paradigm that increased VA-ECMO flow is directly associated with increased LAP. Paradoxically, our results suggest that increased VA-ECMO flow may effectively *reduce* LAP and thereby reduce the risk of pulmonary edema. On the other hand, reducing VA-ECMO speed and flow may exacerbate the sensitivity of LAP to impaired underlying LV contractility, worsening LV filling pressure. These findings are consistent with a recent clinical study conducted in patients supported with VA-ECMO, in which higher VA-ECMO flow was associated with reduced LVEDP and LV end-diastolic volume [18]. In clinical practice, however, these findings need to be balanced against the known effect of increased pump speed on arterial pressure—verified in this study—and the consequent effect on aortic valve opening. In any case, these findings highlight that the role of VA-ECMO speed modulation should be primarily to alter tissue perfusion and aortic valve ejection, *not* to improve left-sided cardiac filling pressure in a simplistic fashion by reducing pump speed.

### 4.6. Limitations

This study was performed in an MCL, which carries a number of inherent limitations, particularly in terms of its ability to faithfully replicate human circulation. Our model may not fully capture the dynamic interactions between the heart, vasculature, and neurohormonal responses that occur in vivo. We used a glycerol/water solution to replicate the viscosity of human blood, although it may differ in terms of flow dynamics, potentially affecting the external validity of our results. Despite these shortcomings, MCLs are advantageous in terms of their ability to facilitate systematic investigation of physiological relationships, while maintaining a physical model that is less susceptible to assumptions than in silico models. We did not specifically assess aortic valve opening in this study. The complex interaction between ECMO flow, aortic valve dynamics, and LV function is poorly understood and warrants further investigation. These results are hypothesis-generating, and need to be verified in additional MCL models, as well as large animal or human hemodynamic studies, before they can be used to directly inform larger clinical trials or clinical practice.

## 5. Conclusions

The effects of VA-ECMO on LAP are largely mitigated in the presence of interventricular interactions and a Frank–Starling mechanism, with a reduction in and stabilization of LAP at the highest VA-ECMO speeds.

## Figures and Tables

**Figure 1 bioengineering-12-00237-f001:**
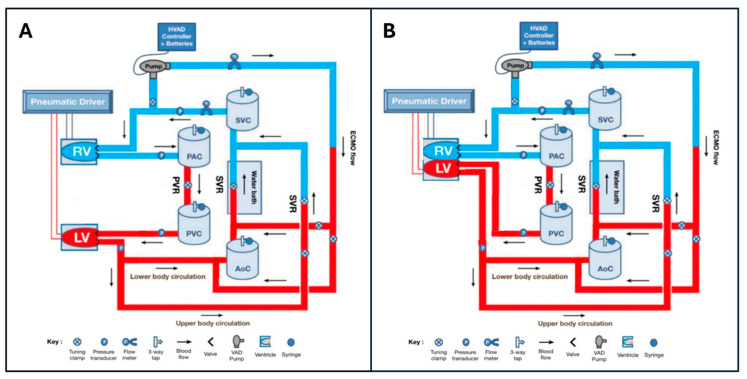
Mock circulatory loop configurations. (**A**): Pre-existing mock circulatory loop in a separate ventricle configuration, adapted from [11]. (**B**): Mock circulatory loop incorporating biventricular heart model. RV = right ventricle, LV = left ventricle, PVC = pulmonary venous compliance, PAC = pulmonary arterial compliance, SVC = systemic venous compliance, AoC = aortic compliance, SVR = systemic venous resistance, PVR = pulmonary vascular resistance, ECMO = extracorporeal membrane oxygenation, HVAD = HeartWare Ventricular Assist Device.

**Figure 2 bioengineering-12-00237-f002:**
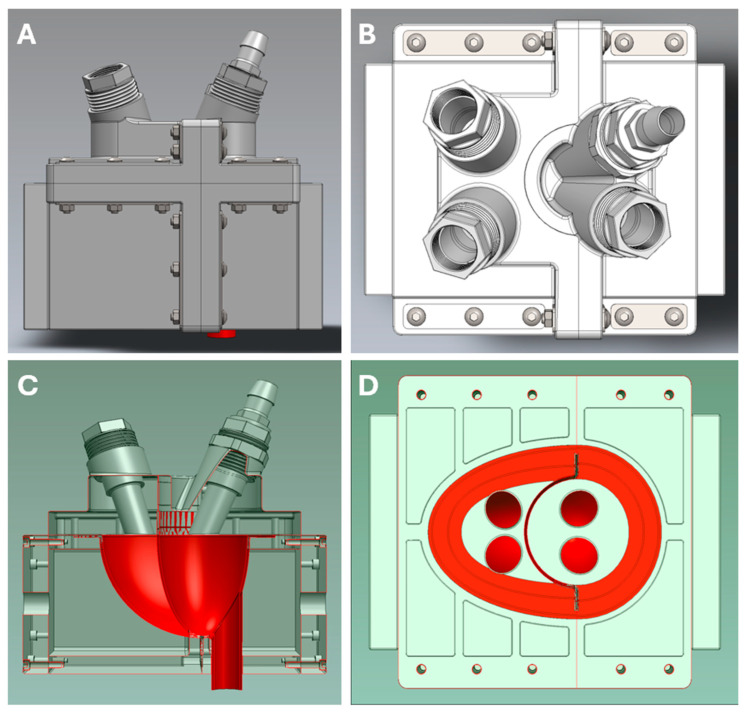
Three-dimensional schematics of biventricular heart box. (**A**): Front view, external. (**B**): Top view, external. (**C**): Front view, internal. (**D**): Top view, internal.

**Figure 3 bioengineering-12-00237-f003:**
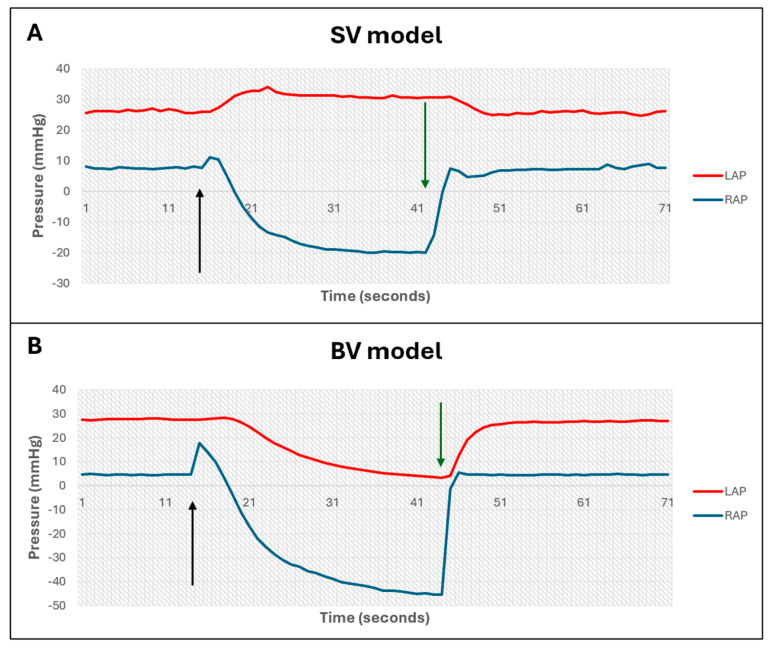
Interventricular interactions following clamping of right atrial inflow. (**A**): Separate ventricle model. (**B**): Biventricular model. Black and green arrows indicate clamping and unclamping of right atrial inflow, respectively. SV = separate ventricle model, BV = biventricular model, LAP = left atrial pressure, RAP = right atrial pressure.

**Figure 4 bioengineering-12-00237-f004:**
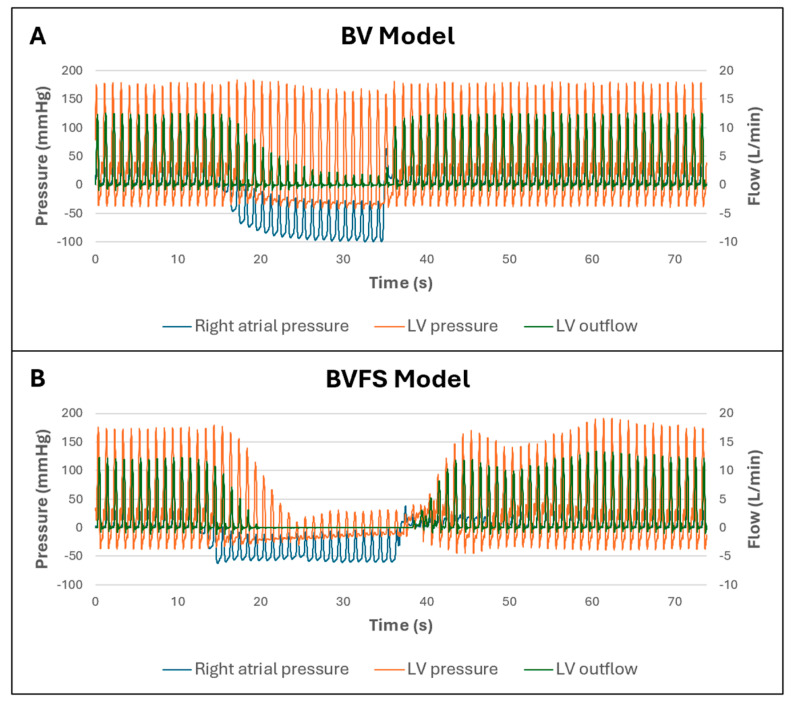
Simulation of Frank–Starling mechanism in the biventricular model via clamping of RA inflow. (**A**): Biventricular model without programmed Frank–Starling mechanism. (**B**): Biventricular model with programmed Frank–Starling mechanism. RAP = right atrial pressure, LVESP = left ventricular end-systolic pressure, LV = left ventricle, BV = biventricular model without Frank–Starling mechanism, BVFS = biventricular model with Frank–Starling mechanism.

**Figure 5 bioengineering-12-00237-f005:**
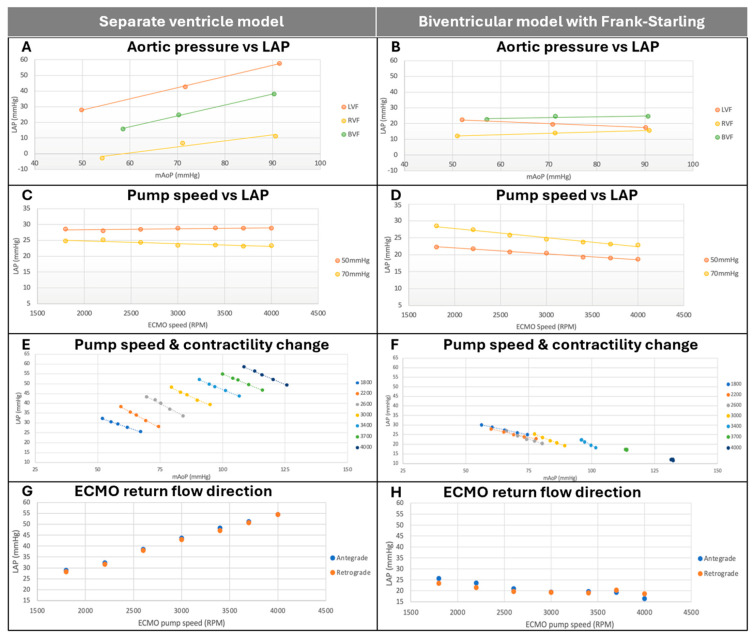
Determinants of left atrial pressure in separate ventricle model (**left**) and biventricular model with Frank–Starling mechanism (**right**). (**A**,**B**): Relationship between aortic pressure and LAP while holding ECMO pump speed constant. (**C**,**D**): Relationship between ECMO pump speed and LAP while holding aortic pressure constant. (**E**,**F**): Effect of changes in ECMO pump speed and LV contractility on aortic pressure and LAP. (**G**,**H**): Effect of ECMO return flow direction on LAP. mAoP = mean aortic pressure, LAP = left atrial pressure, SV = separate ventricle model, BVFS = biventricular model with Frank–Starling mechanism, ECMO = extracorporeal membrane oxygenation, LVF = left ventricular failure, RVF = right ventricular failure, BVF = biventricular failure.

**Table 1 bioengineering-12-00237-t001:** Target values for reproduction of normal heart, LVF, RVF and BVF cardiac states. LVF = left ventricular failure, RVF = right ventricular failure, BVF = biventricular failure, mAoP = mean aortic pressure, LAP = left atrial pressure, RAP = right atrial pressure.

State	mAoP (mmHg)	LAP (mmHg)	RAP (mmHg)	Flow (L/min)
Normal	>60	<15	<10	>4
LVF	>60	>25	<10	<3
RVF	>60	<15	>25	<3
BVF	>60	>25	>25	<3

**Table 2 bioengineering-12-00237-t002:** Reproduction of normal heart, LVF, RVF, and BVF parameters. LVF = left ventricular failure, RVF = right ventricular failure, BVF = biventricular failure, mAoP = mean aortic pressure, LAP = left atrial pressure, RAP = right atrial pressure.

State	mAoP (mmHg)	LAP (mmHg)	RAP (mmHg)	Flow (L/min)
Normal	63.8	12.0	6.4	4.10
LVF	61.7	40.8	3.5	2.44
RVF	62.7	13.2	32.2	2.32
BVF	64.5	32.9	34.0	1.52

## Data Availability

Data will be made available upon reasonable request.

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
