# Peer review of "Increased VA-ECMO Pump Speed Reduces Left Atrial Pressure: Insights from a Novel Biventricular Heart Model"

_bioengineering, 2025, doi:10.3390/bioengineering12030237_

Round 1
Reviewer 1 Report
Comments and Suggestions for Authors
The study by Kasavaraj et al. investigates the effects of veno-arterial extracorporeal membrane oxygenation on left atrial pressure (LAP) using a novel biventricular heart model. The authors aim to understand how interventricular interactions and the Frank-Starling mechanism influence LAP during VA-ECMO support. The study is significant as it addresses a critical gap in understanding the hemodynamic effects of VA-ECMO, particularly in the context of cardiogenic shock, where elevated LAP can lead to pulmonary edema and worsen patient outcomes.
In my opinion, the strength points of the study include the use of a novel heart model (3D printed), with a programmed Frank-Starling mechanism and the ability to replicate normal physiological cardiac states.
The study systematically evaluates the effects of multiple variables on LAP, including ECMO pump speed, mean aortic pressure, LV contractility, and ECMO return flow direction. This comprehensive approach provides a detailed understanding of the complex interactions between these factors.
The inclusion of both separate-ventricle and biventricular models allows for a direct comparison of the effects of interventricular interactions and the Frank-Starling mechanism.
The findings challenge the current clinical paradigm that increased ECMO pump speed necessarily leads to elevated LAP and pulmonary edema. Instead, the study suggests that higher ECMO speeds can reduce LAP.
While the MCL is a valuable tool for studying hemodynamics, it has inherent limitations in replicating the complexity of the human cardiovascular system. The model may not fully capture the dynamic interactions between the heart, vasculature, and neurohormonal responses that occur in vivo. Using a glycerol/water solution to simulate blood may not accurately reflect the properties of human blood, potentially affecting the results, especially in terms of flow dynamics and resistance.
The study does not specifically assess the impact of ECMO on aortic valve opening, which is a critical factor in determining LV afterload and ejection. The complex interaction between ECMO flow, aortic valve dynamics, and LV function warrants further investigation.
The findings are based entirely on an in vitro model, and the study lacks validation in large animal models or human subjects. While the results are promising, they need to be confirmed in clinical settings before they can be applied to patient care.
The study concludes that the direction of ECMO return flow (antegrade vs. retrograde) does not affect LAP. However, this finding contradicts some clinical observations and may be influenced by the simplified nature of the MCL. Further investigation is needed to reconcile these differences.
The study reports statistically significant findings, but the small sample size (n=12) may limit the robustness of these results. Larger studies are needed to confirm these findings.
Minor issues:
The abstract is not structured, with no numerical results.
Reviewer 2 Report
Comments and Suggestions for Authors
Cardiogenic shock with acute hemodynamic decompensation represents the most serious risk and its proper identification may have important implications in terms of pharmacological management, as might procedural planning in case of patients undergoing catheter ablation. Over the last 15 years, clinicians have increasingly used extracorporeal membrane oxygenation (ECMO) as a rescue technique, including cannulating patients in community hospitals without ECMO capabilities, leading to secondary ECMO transports. I congratulate for the very good and technical paper, I have only few minor comments in order to improve the manuscript. I would only open a little section about a more “clinical” application of VA ECMO techniques, including, specifically: acute myocardial infarction treatment, fulminant myocarditis, acute exacerbations of chronic heart failure, cardiac failure due to intractable arrhythmias, post-cardiotomy cardiac failure (DOI: 10.1016/j.hrtlng.2025.01.015). At the same time in this scenario authors should simply cite the importance of provisional vs upfront ECMO strategies. In patients undergoing to catheter ablation, in some intermediate-risk condition, a a pre-defined strategy before VT ablation is crucial. and the use of provisional circulatory support with VA-ECMO during incessant ablation of ventricular arrhythmia may be a safe and winning alternative to upfront strategies (DOI: 10.3390/jcm13154477). Please expand these concepts and cite 2 suggested references
Round 2
Reviewer 2 Report
Comments and Suggestions for Authors
Manuscipt definitely improved. Congratulations to the authors for the very good paper. No further comments